# Anisotropic Zeeman splitting in YbNi$_4$P$_2$

Sara Karbassi[1], Saman Ghannadzadeh[2], Kristin Kliemt[3], Manuel Brando[4],
Cornelius Krellner[3] and Sven Friedemann[1*]

1 H. H. Wills Physics Laboratory, University of Bristol, Bristol, BS8 1TL, UK
2 High Field Magnet Laboratory, University of Radboud, Nijmegen, NL
3 Physikalisches Institut, Goethe-Universität, Frankfurt am Main, Germany
4 Max Planck Institute for Chemical Physics of Solids, Dresden, Germany

* sven.friedemann@bristol.ac.uk

## Abstract

The electronic structure of heavy-fermion materials is highly renormalised at low temperatures with localised moments contributing to the electronic excitation spectrum via the Kondo effect. Thus, heavy-fermion materials are very susceptible to Lifshitz transitions due to the small effective Fermi energy arising on parts of the renormalised Fermi surface. Here, we study Lifshitz transitions that have been discovered in YbNi$_4$P$_2$ in high magnetic fields. We measure the angular dependence of the critical fields necessary to induce a number of Lifshitz transitions and find it to follow a simple Zeeman-shift model with anisotropic $g$-factor. This highlights the coherent nature of the heavy quasiparticles forming a renormalised Fermi surface. We extract information on the orientation of the Fermi surface parts giving rise to the Lifshitz transitions and we determine the anisotropy of the effective $g$-factor to be $\eta \approx 3.8$ in good agreement with the crystal field scheme of YbNi$_4$P$_2$.

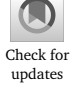

# 1 Introduction

The electronic and magnetic properties of metals are in close relationship with the electronic structure at the Fermi level. For the case of clean metallic ferromagnets at low temperatures, the topology of the Fermi surface has been predicted to be of particular importance, the very existence of a ferromagnetic (FM) quantum critical point (QCP) has been predicted to be absent in two and three-dimensional clean metallic systems due to the coupling of soft fermionic electronic excitations to the magnetic order parameter [1, 2].

A QCP marks a continuous phase transition at zero temperature which experiments usually reveal by tuning a finite-temperature continuous phase transition to zero temperature utilizing a tuning parameter like pressure, composition, or magnetic field. For most ferromagnets the FM QCP is indeed avoided with the transition becoming discontinuous [3, 4] or new phases preempting the FM QCP [5, 6]. Whilst in some cases new quantum critical phenomena like quantum critical end points [7] and quantum tricritical points are found [8], materials with an accessible FM QCP remain of large interest and continue to challenge theoretical concepts. For clean metallic systems in dimensions larger than one, the coupling of fermionic excitations to any uniform magnetisation has been predicted to lead to a discontinuous ferromagnetic transition at low temperatures thus rendering the quantum phase transition discontinuous and removing low-energy quantum fluctuations [1, 9]. This theoretical framework leaves the possibility for one-dimensional systems to promote a FM QCP. Further theoretical scenarios to explain the existence of a FM QCP can arise for heavy-fermion materials like $YbNi_4P_2$ where the disintegration of quasiparticles has been predicted to lead to a FM QCP [10]. Here, we study the electronic structure of heavy-fermion ferromagnet $YbNi_4P_2$ which has been shown to exhibit a FM QCP induced upon small substitution of phosphorus by arsenic [11] and which features one-dimensional chains of ytterbium atoms leading to quasi 1D Fermi surface sheets in unrenormalised band structure calculations [12].

The stoichiometric parent compound $YbNi_4P_2$ exhibits ferromagnetic order at a very low ordering temperature of $T_C = 0.17\,\text{K}$ from a heavy-fermion state with hallmarks of strong hybridisation between trivalent ytterbium magnetic moments and conduction electrons. Coherent heavy quasiparticles and a heavily renormalised band structure are present below the Kondo-lattice temperature $T_K \approx 8\,\text{K}$. This Kondo-lattice effect gives rise to a large density of states at the Fermi level. For many heavy-fermion systems the Fermi surface is very similar in shape to the underlying non-renormalised Fermi surface whilst the Fermi volume increases. For $YbNi_4P_2$ one can expect some reminiscence of the quasi-1D Fermi surface character in the renormalised state.

The large density of states and concomitant reduced dispersion at the Fermi level promote a large response of the Fermi surface topology to an external magnetic field as the Zeeman energy $1/2g_{eff}\mu_B B$ can reach values comparable to the Kondo energy at accessible fields. Thus, numerous Lifshitz transitions (LTs) are a strong signature of a Fermi surface of heavy quasiparticles as indeed observed in $YbNi_4P_2$ as well as $YbRh_2Si_2$ [13–15]. In $YbNi_4P_2$, a set of nine LTs has been identified through measurements of magnetoresistance, thermopower, magnetisation, and magnetostriction [15]. Here, we analyse the angular dependence of the magnetic field necessary to induce the LTs in $YbNi_4P_2$ and extract the anisotropy of the effective $g$-factor. Finally, we infer the orientation of the neck-type LTs and identify parts of the Fermi surface that match this orientation as candidates for the LTs.

A LT marks a change of topology of the Fermi surface which can be induced by Zeeman splitting of an electronic band e.g. when the minority spin Fermi surface vanishes or a Fermi surface emerges as the majority spin band becomes populated, both these cases are called void-type. Other LTs include the formation or breaking of a neck-like link between disconnected parts of the Fermi surface. A Zeeman-splitting induced LT occurs when the Zeeman energy

$\frac{1}{2}g_{\text{eff}}\mu_{\text{B}}B$ becomes larger than the effective Fermi energy $\tilde{E}_{\text{F}}$ of the relevant part of the Fermi surface. For some parts of the Fermi surface, $\tilde{E}_{\text{F}}$ can be small, e.g. small pockets or narrow links. Even without the knowledge of the effective Fermi energy, measurements of the magnetic field at which a LT occurs provides valuable information. For instance, the angular dependence of the effective $g$-factor can be inferred as long as the Zeeman shift of rigid bands is sufficient to describe the field induced changes to the electronic band structure for different orientations of the magnetic field.

In general, the $g$-factor is a tensor. For most tetragonal systems, however, only two components are relevant: $g_{\perp}$ and $g_{\parallel}$, the effective $g$-factor for field perpendicular and parallel to the crystallographic direction (001). For a polar angle $\phi$ the effective $g$-factor is given by a linear combination $g_{\text{eff}}(\phi) = \sqrt{g_{\parallel}^2 \cos^2\phi + g_{\perp}^2 \sin^2\phi}$. Thus, the critical magnetic field $B_n$ for a LT will vary as

$$B_n = A_n \left[ g_{\parallel}^2 \cos^2(\phi) + g_{\perp}^2 \sin^2(\phi) \right]^{-1/2}, \tag{1}$$

where the factor $A_n$ includes the unknown $\tilde{E}_{\text{F}}$. For YbNi$_4$P$_2$, the situation is slightly more complicated as Yb ions occupy sites with orthorhombic symmetry. Two identical sites which are oriented 90° relative to each other exist along (110) and (1-10). We choose the field orientation such that the two sites experience the same symmetry (including the direction of the magnetic field). This is achieved for magnetic field in the (001)-(100) plane.

## 2 Experimental Details

Magnetoresistance measurements were used to track the Lifshitz transitions in YbNi$_4$P$_2$ as a function of the orientation of magnetic field. The magnetic field was rotated between the crystallographic directions (001) and (100). Two samples were cut to measure the c-axis resistivity $\rho_{zz}$ and the in-plane resistivity $\rho_{xx}$ as sketched in Fig. 1. For c-axis magnetoresistance ($\rho_{zz}$), the configuration is changing from longitudinal to transverse as the field rotates away from (001). For the in-plane magnetoresistance ($\rho_{xx}$), the magnetic field is rotating in the plane spanned by (001) and (010) and thus is always oriented transverse to the current.

Measurements were conducted at a temperature of $T \approx 0.4\,\text{K}$ in a Bitter magnet up to 30 T at the Nijmegen High Magnetic Field Laboratory using a rotator where the orientation was monitored in situ with a Hall sensor. The resistance was measured using standard 4-point lock-in method with a typical current of 3 mA where heating was found to be negligible.

The single crystals of YbNi$_4$P$_2$ were grown from levitating melt using the Czochralski method [16]. Samples were cut from oriented ingots and polished to thin, long platelets. Contacts were spot welded and mechanically stabilised with silver epoxy. Our samples have resistance ratios between 300 K and 2 K in excess of 15.

## 3 Computational Details

The full potential augmented plane wave and local orbitals WIEN2k density functional theory code was used to perform band structure calculations [17]. Band energies were calculated in the first Brillouin zone on a 1 000 000 $k$-point mesh. The Perdew-Burke-Enzerhof generalised gradient was used as an approximation to the exchange correlation potential. The band structure was calculated using the lattice constants experimentally determined for YbNi$_4$P$_2$ [12]. The $f$-core state was simulated by performing the calculation for LuNi$_4$P$_2$ (using the YbNi$_4$P$_2$ lattice constants). The energy range for valence states includes $-6$ Ry to 5 Ry corresponding to a valence band treatment of the $4f$, $5s$, $5p$, $5d$, and $6s$ electrons for Lu, the $3d$ and $4s$

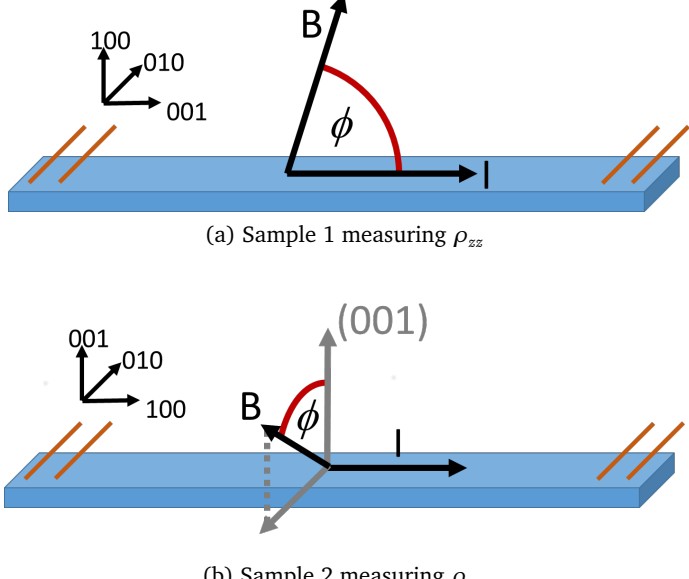

(a) Sample 1 measuring $\rho_{zz}$

(b) Sample 2 measuring $\rho_{xx}$

Figure 1: Current and magnetic field orientations for c-axis and in-plane magnetoresistance measurements.

electrons for Ni, and the $3s$ and $3p$ electrons for P. This resulted in more than 99.5 % of the Lu $4f$ states localised to the core. Relativistic effects and spin-orbit coupling were included on a one-electron level for Lu and Ni while P have been approximated as non-relativistic owing to its light mass. Our band structure and Fermi surface topology matches that of earlier $f$-core calculations [12].

## 4 Results and Discussion

The magnetoresistance $\rho_{zz}$ is shown in Fig. 2 for selected polar angles $\phi$. The excellent sample quality is evident from quantum oscillations present at magnetic fields above 20 T which are being studied in more detail and will be presented in a future publication. The curve with $B \parallel (001)$ corresponds to the data analysed in Ref. [13] where it was shown that the kinks in magnetoresistance are present at many of the LTs identified via a combination of thermopower, magnetostriction, magnetisation, and magnetoresistance measurements. We follow the scheme of Ref. [13] in labelling the LTs $B_n$ with $n$ ranging from 1 to 9. Also, it was shown that the positions of the LTs do not shift in the temperature range between 0.03 K to 0.65 K indicative of coherent quasiparticles with a fully established Fermi surface subject to the Zeeman effect.

As the magnetic field is rotated away from the (001) direction, the features identified with the LTs shift to higher fields. In Fig. 2(b) we show two methods employed to track their position as a kink in the magnetoresistance (e.g. $B_9$) or a step in its first derivative (e.g. $B_3$). In both cases a constant offset was subtracted. These two methods yield values $B_n$ in good agreement with each other but with different levels of uncertainty depending on the background. We use the position of the step in the first derivative to extract $B_3$ and the position of the kink in magnetoresistance to extract $B_4$, $B_7$, $B_8$, and $B_9$.

The signatures are broadened significantly for polar angles in excess of 45° leading to increased uncertainty in their identification. Above 70° most of the signatures cannot be identified unambiguously.

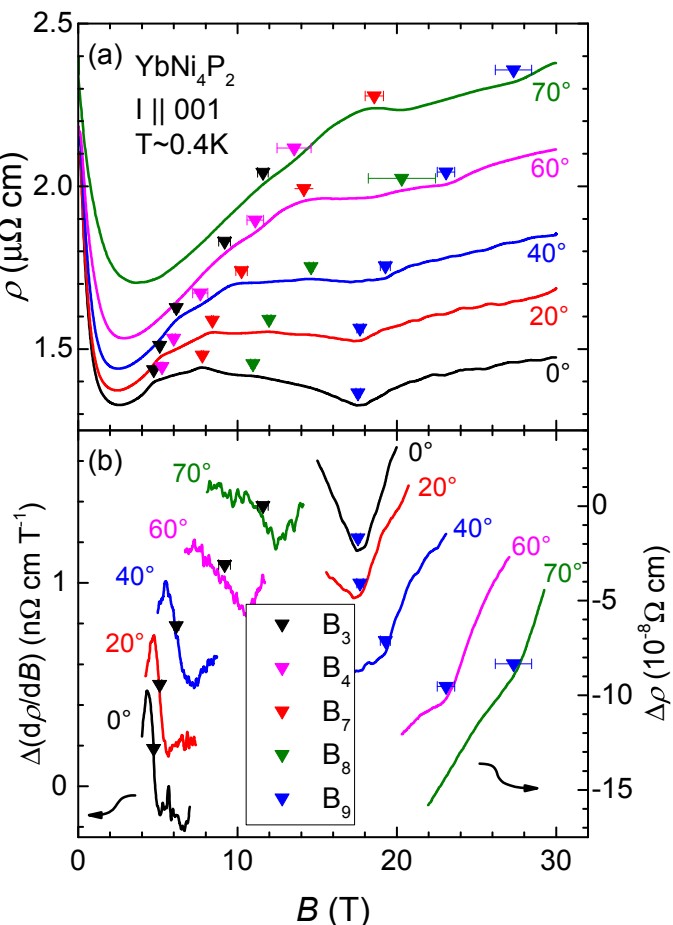

Figure 2: (a) $c$-axis magnetoresistance $\rho_{zz}$ for representative orientations. Labels indicate the polar angle $\phi$ between the magnetic field and the crystallographic (001) direction. Arrows indicate the position of Lifshitz transitions identified from the derivative and constant background subtracted signal as shown in (b) and as described in the text. Data in (a) and (b) are offset for clarity.

Fig. 3 shows the angular dependence of $B_n$ for the LTs that can be identified in magnetoresistance. We were able to track the position of $B_3$, $B_4$, $B_7$, $B_8$, and $B_9$. For all except $B_9$ we obtain a very good description of the angular dependence using eq. (1) (solid red lines in Fig. 3). This confirms that the underlying LTs of $B_3$, $B_4$, $B_7$, and $B_8$ arise from the Zeeman effect acting on the fully established Fermi surface of coherent quasiparticles that can be modelled by a rigid band shift.

As the factor $A_n$ in eq. 1 is dependent on the unknown effective Fermi energy for each LT we cannot obtain absolute values for the $g$-factor. However, we can determine the anisotropy $\eta$ of the $g$-factor

$$\eta = \frac{g_{\|}}{g_{\perp}}, \tag{2}$$

which is independent of $A_n$. The values obtained for the anisotropy of each LT are listed in Tab. 1. For the LTs extracted from $\rho_{zz}$, the values for $\eta$ agree within experimental uncertainty in the range from 3.5 to 3.9 for all LTs except $B_9$ where the best fit to the data does not provide a confident prediction for the anisotropy.

The LT at $B_9$ shows strong discrepancy with the form of eq. 1. Both the slower increase

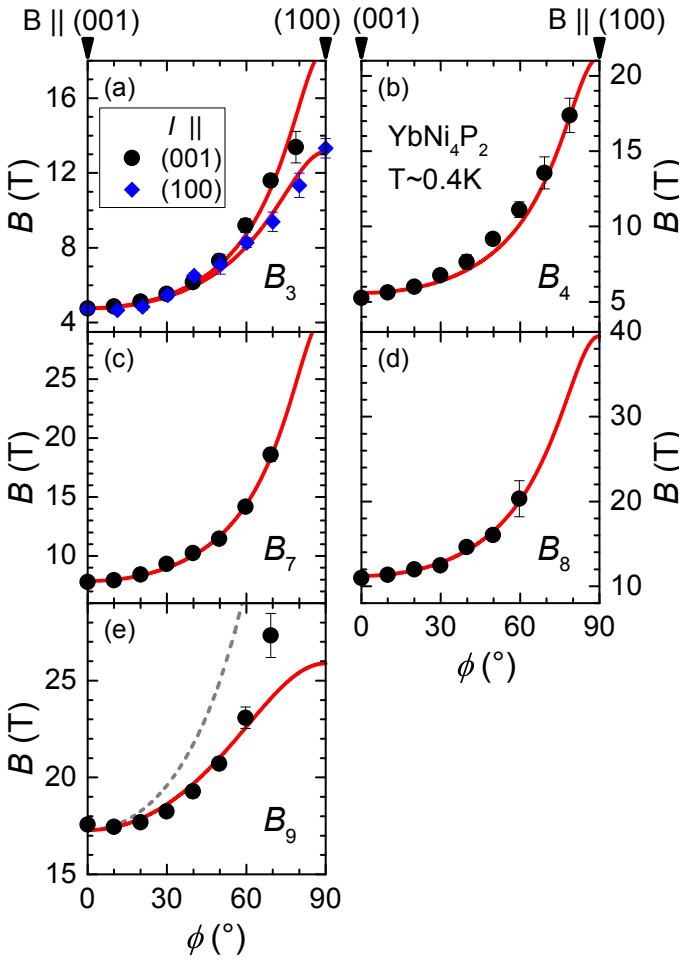

Figure 3: Magnetic anisotropy in YbNi$_4$P$_2$. (a)-(e) The position of the LTs identified from $\rho_{zz}$ (black circles) (cf. Fig. 2) and $\rho_{xx}$ (blue diamonds) (cf. Fig. 4) are presented together with uncertainty weighted least square fits using eq. 1 (red solid lines). For $B_9$ only a poor fit could be obtained as shown by the red line in (e) corresponding to $\eta = 1.5$. The dashed grey line in (e) represents the expected angular dependence for $\eta = 3$ fixed to the position of $B_9$ at $\phi = 0$.

at low angles $\phi \leq 30°$ and the steeper increase for $\phi > 50°$ make it impossible to obtain a good fit of $B_9(\phi)$. The anisotropy of $B_9$ seems to be significantly smaller than for the other LTs. This is apparent from the comparison with the form of eq. 1 for $\eta = 3$ included as a dashed line in Fig. 3(e). We conclude that the LT at $B_9$ cannot be described by a rigid band shift only

Table 1: Characteristics of the LTs: magnetic field values $B_c = B_n(\phi = 0)$ as identified in Figs. 2 and 4 as well as the $g$-factor anisotropy $\eta$ extracted from fits to $B_n(\phi)$ as shown in Fig. 3 for $\rho_{zz}$ and $\rho_{xx}$. Errors denote standard errors from the least-squares fits.

|  | $\rho_{zz}$ | | | | $\rho_{xx}$ |
|---|---|---|---|---|---|
|  | $B_3$ | $B_4$ | $B_7$ | $B_8$ | $B_3$ |
| $B_c$ (T) | 4.8(1) | 5.1(1) | 7.8(1) | 11.0(1) | 4.7(1) |
| $\eta$ | 3.9(3) | 3.8(4) | 3.9(3) | 3.5(5) | 2.8(2) |

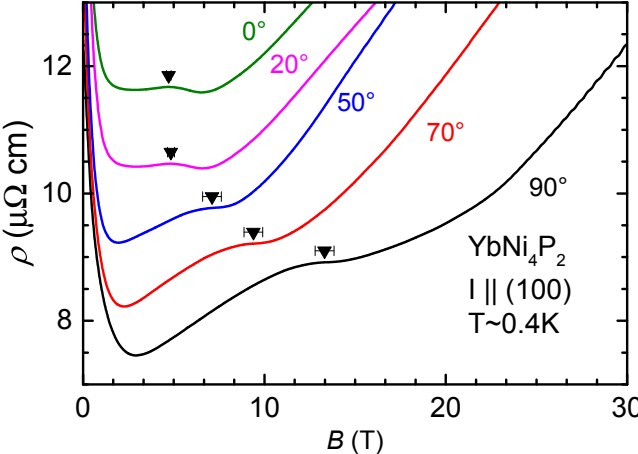

Figure 4: (a) in-plane magnetoresistance $\rho_{xx}$ for representative orientations. Labels indicate the polar angle $\phi$ between the magnetic field and the crystallographic (001) direction. Arrows indicate the position of representative Lifshitz transitions. Data are offset for clarity.

but rather points towards a strong suppression of the Kondo effect [18] that varies over the field and angular range of this LT. The suppression of the Kondo effect will lead to a reduced dispersion and thus could lead to an accelerated increase of $B_9$ as observed for $\phi > 50°$. In fact, among the cascade of LTs $B_9$ occurs at the largest field where the Zeeman energy becomes larger than the Kondo temperature.

In Fig. 4 the in-plane magnetoresistance $\rho_{xx}$ is presented. Here, we are only able to unambiguously identify $B_3$ as a kink-like maximum in $\rho(B)$ for $B \parallel (001)$ ($\phi = 0$) which develops into a shoulder for large polar angles $\phi \geq 50°$. The angular dependence of $B_3$ from $\rho_{xx}$ is included in Fig. 3(a) (blue diamonds) and can be well described by eq. 1 with an anisotropy $\eta = 2.8(2)$ somewhat smaller than that determined from $\rho_{zz}$.

All LTs except $B_9$ detected here have been identified to be of neck type [13]. The magnetoresistance from a neck can be described by the same method used for two-dimensional Fermi surface sheets [19] which predicts strong magnetoresistance for the current direction along the neck axis only. Thus, the fact that almost all LTs are observed for the current along the (001) direction points towards the necks to be oriented along (001). This could for instance be narrow, tube-like links as suggested by DFT calculations (cf. Fig. 5). Here, a neck may form by the separate parts joining in agreement with the analysis of the thermopower [13].

For magnetic field orientations starting from $\phi = 0$ such neck-type links have small, closed orbits as highlighted in Fig. 5. In fact, some closed orbits will be arbitrarily small for $B \leq B_n$ before the neck is joined. Such small orbits will experience a strong magnetoresistance even at low fields as the high-field condition $\omega_c \tau > 1$ with the cyclotron frequency $\omega_c$ and the scattering time $\tau$ can be easily satisfied. We note that these orbits are not expected to yield quantum oscillations as these are not extremal orbits for fields $B \leq B_n$ when the neck is not joined.

For field orientations close to $\phi = 90°$ the magnetoresistance contribution to $\rho_{zz}$ of necks oriented along (001) will be reduced as only open orbits traversing the sheets and running into and out of the neck pits will be present at fields $B \leq B_n$. Open orbits are much less efficient at averaging momentum and thus generate much weaker magnetoresistance. This is consistent with the observation of weaker signatures for $B_3$, $B_4$, $B_7$, $B_8$, and $B_9$ at larger angles $\phi$. Thus, the fact that the signatures in $\rho_{zz}$ become weaker as the field orientation approaches (100) is consistent with necks oriented along the (001) direction.

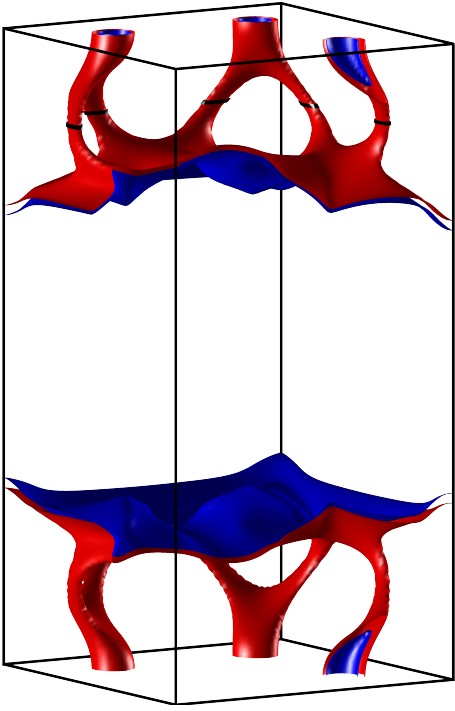

Figure 5: Topology of one of the $f$-core Fermi surface sheets obtained from uncorrelated DFT calculations. This sheet is displayed in blue for the Fermi energy obtained by the DFT calculations. Necks are connected in the $f$-core DFT calculations if the Fermi energy is shifted by $-1.5$ meV. Electron orbits at the neck are highlighted in black.

Only the LT at $B_3$ is observed in $\rho_{xx}$ even though it has been identified to be of neck type just like $B_4$, $B_7$, and $B_8$. At the same time a much reduced anisotropy is extracted for $B_3$ from $\rho_{xx}$. This leaves some doubts about the association of the signature in $\rho_{xx}$ with a LT at $B_3$. Further measurements of the thermopower, magnetostriction, and temperature dependence of the magnetoresistance will hopefully clarify the nature of the signature at $B_3$ in $\rho_{xx}$.

The overview of the anisotropies in Tab. 1 reveals significantly larger values for $\rho_{zz}$ compared to $\rho_{xx}$. Here, we restrict the discussion to all LTs except $B_9$ which was identified above to require modelling beyond the rigid band shift underlying eq. 1. All values of $\eta$ from $\rho_{zz}$ agree within experimental uncertainty with a mean of 3.8(1). This value agrees well with the expected $g$-factor anisotropy $\eta_{\mathrm{CEF}} = 3.93$ of the Yb crystal-field ground state [20]. The average including the anisotropy obtained from $\rho_{xx}$ amounts to $\eta = 3.1(2)$ which is in less good agreement with $\eta_{\mathrm{CEF}}$. This reduced agreement of $\eta$ extracted from $\rho_{xx}$ provides further evidence that the signature in $\rho_{xx}$ is not arising from the LT at $B_3$. Thus, we conclude the $g$-factor anisotropy is most likely given by the mean extracted from $\rho_{zz}$ measurements as $\eta \approx 3.8$.

## 5 Conclusions

In summary, we present the angular dependence of the magnetic field where Lifshitz transitions are induced in YbNi$_4$P$_2$. Our measurements confirm the easy magnetic direction of the fluctuating magnetic moments above $T_{\mathrm{C}}$ to point along the crystallographic (001) direction. The extracted strong anisotropy of the effective $g$-factor suggests a correlated electronic band structure in which the Yb $f$ state in the anisotropic crystal field ground state hybridizes with the conduction electrons. This is similar to other Yb heavy-fermion systems [21, 22]. We find

indications for the orientation of the neck-type LTs at $B_3$, $B_4$, $B_7$, and $B_8$ to form close to the (001) direction in $k$-space. The void-type LT at $B_9$ cannot be described by a rigid band shift model and rather suggests a partial breakdown of the Kondo effect at this large magnetic field. This knowledge of the angular dependence of LTs is expected to be important for the interpretation of future quantum oscillation measurements as demonstrated for other heavy-fermion systems [23, 24] as well as other field- and angular dependent measurements.

# Acknowledgements

The authors would like to thank Oliver Stockert and Christoph Geibel for valuable discussion. This work was partially supported by the EPSRC under grant EP/N01085X/1 and DFG projects KR3831/4-1 and BR4110/1-1.

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
