# Peer review of "Anisotropic Zeeman Splitting in YbNi4P2"

_SciPost Physics, doi:SciPost Phys. 5, 056 (2018)_

## Round 1 · Referee Report · Anonymous (Referee 4) · 2018-9-30

Strengths
1- Provides useful follow-up on a recent PRL
2- Gives interesting information about the evolution of the Fermi surface of YbNi4P2 as a function of an applied magnetic field
Weaknesses
1- The abstract and the introduction contain numerous misleading and/or confusing statements.
Report
This paper provides a follow-up to a recent PRL by a similar set of authors (Ref. [13] in the MS). Specifically, it discusses additional evidence from transport data for the occurrence and locations of the nine Lifshitz transitions that were identified in Ref. [13]. It also puts observations in context by pointing out the very unusual nature of the material under consideration.
The only weakness of the paper is a series of confusing and/or misleading statements in the contextual parts, starting with the abstract. I will list these in the order in which they occur.
1) Abstract, 1st sentence: This statement is true only for clean metallic systems. In fact, Ref. [2] discusses many examples of continuous FM zero-temperature transitions in systems that contain substantial amounts of quenched disorder.
2) Abstract, 2nd sentence: As written, this implies that the prediction in question is made in the current paper. This does not seem to be the case, so this should probably read "The electronic structure has been predicted ..."
3) p.1, 1st paragraph, 4 lines down: "... is predicted to be absent ..." has the same problem: it should read "... has been predicted ..." and contain a qualifier saying that this applies to clean systems only.
4) p.1, 2nd paragraph, 2 lines down: "... suppressing a ... transition ..." The authors clearly don't mean to say that the transition is being suppressed. This needs to be rephrased as "tuning the transition temperature to zero", or something similar.
5) p.2, 2nd paragraph, 3rd line from bottom "... the topology of the Fermi surface changes slightly ...". A `slight change of topology' is something of an oxymoron. At the very least the authors need to explain what they mean by this.
6) p.2, 3rd paragraph: "Here, we provide further evidence ..." It is not clear to the referee what the authors mean to say. Do they mean "we report on signatures in the magnetoresistance that reflect LTs"? Or, "we show that signatures in the magnetoresistance can provide evidence for LTs"? Or something else?
7) p.2, just before Eq. (1): "the magnetic field of a LT" is a strange formulation. "the critical field for a LT", or "the field at which a LT occurs" would be clearer.
A few minor linguistic problems:
8) p.1, last line: "gapping" is misspelled.
9) p.2, 2nd paragraph: In the sentence "This Kondo-lattice effect ..." there is an "a" too many.
9) p.3, 2d paragraph of Sec. 2: "home-build rotator" should be "custom-built rotator" or something similar.
Requested changes
Fix the numbered issues listed in the report above.

---

## Round 1 · Referee Report · Anonymous (Referee 3) · 2018-10-11

Strengths
1. Deals with interesting Yb compound close to a ferromagnetic quantum critical point.
2. Describes an angular magnetoresistance study to extract positions of Lifshitz transitions.
3. Anisotropy of the effective g-factor has been determined
Weaknesses
1. We do not learn much more about the interesting physics of YbNi4P2.
2. Quantum oscillations in the data are not analysed.
3. Fermi surface calculations and discussion are fragmentary.
Report
Karbassi et al. have performed magnetoresistance measurements on the heavy-fermion material YbNi4P2 in two different configurations: c-axis and in-plane. This material attracts attention because it is close to a FM QCP. As function of the field a number of Lifshitz transitions have been observed (Ref.13). In this paper the authors study their angular dependence. The data are used to extract the anisotropy of the effective g-factor. This is the main result.
The data are sound and the analysis is to the point, but even with these new results we do not learn much more about YbNi4P2 and FM quantum criticality. An important feature in the data is the observation of quantum oscillations, but these will be reported elsewhere… The authors show one sheet of the probably complex Fermi surface (Fig.5) and indicate neck positions, but proof therefore is lacking. The authors made an angular dependent study of quantum oscillations, which should be explored to support (or not) their idea.
Other concerns:
1. The authors mention they have a high-quality single crystal but the RRR is only 15. Indeed quantum oscillations are visible in high fields only.
2. Lifshitz transitions B4 and B8 are tabulated, but it is not described how the magnetic field values are determined from the data.
3. The information about the band structure calculations and Fermi surface is fragmentary and insufficient. Probably a full calculation with several Fermi surface sheets has been made, since the authors write “one of the f-core Fermi surface sheets”. It is ok to present only one sheet, but then a reference should be given to the full calculation.
In summary, this work presents only an incremental advance in the understanding of YbNi4P2 and as such is not strong enough for publication in SciPost. I recommend to extend the paper by an analysis of quantum oscillations and compare the Fermi surface cross sections with the calculated Fermi surface at the supposed necking points.
Requested changes
1. See concerns above.
2. Extend the paper to come to a more complete analysis using the available information

---

## Round 2 · Referee Report · Anonymous (Referee 1) · 2018-11-8

Report

I have read the amended paper and am satisfied with the changes made and with the authors' responses. I recommend publication of the current version as is.

---

## Round 2 · Referee Report · Anonymous (Referee 2) · 2018-11-15

Strengths

see previous report

Weaknesses

see previous report

Report

The authors have improved the paper by including a section on electronic strcuture calculations. However, they did not expand it with an analysis of the SdH oscillations. Thus the impact of the paper will remain weak.

---

## Round 2 · Author Response

Reply to Report #1
We thank the referee for his recommendation and his very useful comments. We are glad to see that the referee considers our manuscript as “useful” and “interesting”. We accept the suggested changes to the wording in the manuscript and have included these in our resubmission. In particular, we have clarified the wording to ensure that our discussion is related to clean metallic systems only.

Reply to Report #2
We thank the referee for his time and detailed response. The referee highlights that YbNi4P2 is an interesting compound and that our “main result ..[ is the magnetic] anisotropy and effective g-factor”. This result is novel and will help understand YbNi4P2 in future studies of magnetic and electronic properties. We highlight in our manuscript that the insight will be important for the interpretation of quantum oscillation studies.

“The data are sound and the analysis is to the point, but even with these new results we do not learn much more about YbNi4P2 and FM quantum criticality”

We thank the referee for highlighting the quality of our measurements and analysis. We agree that the immediate insight for the FM quantum criticality is limited but expect that it will be important for the analysis of future measurements like quantum oscillations which will be directly relevant for understanding the FM quantum criticality. Given this, we have revised the first two sentences of the abstract to reflect the focus on the Lifshitz transitions.

“An important feature in the data is the observation of quantum oscillations, but these will be reported elsewhere… The authors show one sheet of the probably complex Fermi surface (Fig.5) and indicate neck positions, but proof therefore is lacking. The authors made an angular dependent study of quantum oscillations, which should be explored to support (or not) their idea.”
“In summary, this work presents only an incremental advance in the understanding of YbNi4P2 and as such is not strong enough for publication in SciPost. I recommend to extend the paper by an analysis of quantum oscillations and compare the Fermi surface cross sections with the calculated Fermi surface at the supposed necking points.”

The referee criticises that quantum oscillations visible in our data at high field have not been fully analysed. We agree that a full study of quantum oscillations will be interesting and should shed further light on the electronic structure of YbNi4P2. However, our quantum oscillation study is not complete. Further measurements at international facilities have been scheduled for 2019 and we are working with colleagues to calculate the band structure of YbNi4P2 taking into account the renormalisation due to the Kondo effect which will be needed for a meaningful comparison with the complex angular dependence of the quantum oscillation frequencies we have found so far. We expect that these results will be available at the end of 2019 / beginning 2020.
The full analysis of the quantum oscillations is expected to be a large manuscript with 10 figures and 5 tables. Including this in the present manuscript would lead to a approximately 4 times larger a manuscript. We believe the main result of the present manuscript (g-factor anisotropy) would not be recognised as part of such an extensive manuscript.
More specifically, the data presented in this manuscript focuses on one rotation of the magnetic field, i.e. from (001) to (100) which is uniquely suited to study the g-factor anisotropy as described in the manuscript. For a full quantum oscillation study, rotations in other directions will be needed, e.g. rotation from (001) to (110) and (100) to (110). These measurements will need to be done at high magnetic field facilities as quantum oscillations are only present above 20 T (as noted by the referee).
Finally, we would like to emphasise that we obtain evidence for the orientation of the neck-type Lifshitz transitions from the analysis of the angular dependence of the critical field and the signatures for different current directions, i.e. purely from our experimental observations of the Lifshitz transitions. The band structure calculations allow us to identify candidates for these Lifshitz transitions. Thus, our conclusions are not dependent on the band structure calculations and quantum oscillation measurements.

---

## Round 2 · List of Changes

Point-by-point changes in response to report #1

1) Abstract, 1st sentence: This statement is true only for clean metallic systems. In fact, Ref. [2] discusses many examples of continuous FM zero-temperature transitions in systems that contain substantial amounts of quenched disorder.

We have clarified throughout the manuscript that our discussion relates to clean metallic systems.

2) Abstract, 2nd sentence: As written, this implies that the prediction in question is made in the current paper. This does not seem to be the case, so this should probably read "The electronic structure has been predicted ..."

We have corrected the wording throughout the manuscript to clarify that we relate to predictions from the papers cited.

3) p.1, 1st paragraph, 4 lines down: "... is predicted to be absent ..." has the same problem: it should read "... has been predicted ..." and contain a qualifier saying that this applies to clean systems only.

Corrected as detailed in 1) and 2)

4) p.1, 2nd paragraph, 2 lines down: "... suppressing a ... transition ..." The authors clearly don't mean to say that the transition is being suppressed. This needs to be rephrased as "tuning the transition temperature to zero", or something similar

Corrected as suggested by the referee

5) p.2, 2nd paragraph, 3rd line from bottom "... the topology of the Fermi surface changes slightly ...". A `slight change of topology' is something of an oxymoron. At the very least the authors need to explain what they mean by this.

We have clarified this statement. The shape of the Fermi surface is often found to be preserved whilst the volume is changed.

6) p.2, 3rd paragraph: "Here, we provide further evidence ..." It is not clear to the referee what the authors mean to say. Do they mean "we report on signatures in the magnetoresistance that reflect LTs"? Or, "we show that signatures in the magnetoresistance can provide evidence for LTs"? Or something else?

We have omitted this sentence as it is confusing. Rather we clarify that the LTs have previously been identified in magnetoresistance in YbNi4P2.

7) p.2, just before Eq. (1): "the magnetic field of a LT" is a strange formulation. "the critical field for a LT", or "the field at which a LT occurs" would be clearer.

We have rephrased as suggested.

A few minor linguistic problems:
8) p.1, last line: "gapping" is misspelled.

Rephrased as linguistics of verb gap aren’t clear to us

9) p.2, 2nd paragraph: In the sentence "This Kondo-lattice effect ..." there is an "a" too many.

Corrected as requested.

10) p.3, 2d paragraph of Sec. 2: "home-build rotator" should be "custom-built rotator" or something similar.

Omitted as the specifications of the rotator are described in text.

Point-by-point changes in response to report #2

We rephrase our manuscript to indicate that the quantum oscillation measurements will require further measurements.

1. The authors mention they have a high-quality single crystal but the RRR is only 15. Indeed quantum oscillations are visible in high fields only.

We would like to emphasise that the resistance ratio at 2K was quoted as 15. The resistance ratio at 0.3K is in excess of 30 and suggests a residual resistance ratio of 40 for our samples. We have quoted the resistance ratio at 2K as this is most convenient for comparisons in absence of a measurement below 0.05K. In addition, we would like to emphasise that the residual resistance ratio of different materials is difficult to compare. The fact that quantum oscillations are visible shows that the crystal quality is high. Our preliminary analysis of quantum oscillations yields a mean free path of 60 nm suggesting a combined concentration of impurities and dislocations of less than 2ppm. In the absence of a standard for crystal quality criteria we consider this high quality.

2. Lifshitz transitions B4 and B8 are tabulated, but it is not described how the magnetic field values are determined from the data.

The Lifshitz transitions B4 and B8 have been determined as kinks in the data as described in the original manuscript. We have revised Fig2 to include the positions of B4 and B8. In addition, we have added an explanation to the text for each field how they have been extracted.

3. The information about the band structure calculations and Fermi surface is fragmentary and insufficient. Probably a full calculation with several Fermi surface sheets has been made, since the authors write “one of the f-core Fermi surface sheets”. It is ok to present only one sheet, but then a reference should be given to the full calculation.

We include details of the calculations in the revised manuscript in section 3.

---

## Editorial Decision

published